# ANYBIMANUAL: TRANSFERRING SINGLE-ARM POLICY FOR GENERAL BIMANUAL MANIPULATION

## ABSTRACT

Performing language-conditioned bimanual manipulation tasks is of great importance for many applications ranging from household service to industrial assembly. However, teleoperating dual-arm demonstrations is expensive due to the high-dimensional action space, which poses challenges for conventional methods to handle general bimanual manipulation tasks. In contrast, single-arm policy has recently demonstrated impressive generalizability across a wide range of tasks because of scaled model parameters and training data, which can provide sharable manipulation knowledge for dual-arm systems. To this end, we propose a plug-and-play method named **AnyBimanual**, which transfers pretrained single-arm policy to multi-task bimanual manipulation policy with limited bimanual demonstrations. Specifically, we first introduce a skill manager to dynamically schedule the discovered skill primitives from pretrained single-arm policy for bimanual manipulation tasks, which combines skill primitives with embodiment-specific compensation. To mitigate the observation discrepancy between single-arm and dual-arm systems, we present a voxel editor to generate spatial soft masks for visual embedding of the workspace, which aims to align visual input of single-arm policy model for each arm with those during pretraining stage. Extensive results on 13 simulated and real-world tasks indicate the superiority of AnyBimanual with an improvement of 12.67% on average success rate compared with previous state-of-the-art methods.

## 1 INTRODUCTION

Dual-arm systems play an important role in robotic manipulation due to the high capacity of completing diverse tasks in household service (Zhang et al., 2024), robotic surgery (Hu et al., 2023) , and component assembly in factories (Buhl et al., 2019). Compared to single-arm systems, dual-arm systems enlarge the workspace and are able to handle more complex manipulation tasks by stabilizing the target with one arm and interact with that using another arm (Grannen et al., 2023; Liu et al., 2024). Even for the tasks that single-arm policies can handle, dual-arm systems are often more efficient because multiple action steps can be simultaneously accomplished (Grotz et al., 2024). Since modern robotic applications require the robot to interact with different tasks and objects, it is desirable to design a generalizable policy model for bimanual manipulation.

To enhance the generalization ability of the manipulation agent, prior works present to leverage the high-level reasoning and semantic understanding capabilities of foundation models like Large Language Models (LLMs) and Vision Language Models (VLMs) to breakdown tasks into executable sub-tasks that can be solved by external low-level controllers (Huang et al., 2023; Gao et al., 2024b; Liu et al., 2023; Joublin et al., 2023; Gbagbe et al., 2024), which thus struggles with contact-rich tasks that requires complex and precised low-level motion. To generalize to contact-rich tasks, recent methods (Kim et al., 2024; Team et al., 2024; Ahn & et al., 2024) tend to learn robotic foundation models directly from large-scale teleoperation data (Khazatsky & et al., 2024; Collaboration & et al., 2024; Walke et al., 2024), which has shown impressive generalizability across various single-arm tasks. However, dual-arm demonstrations are extremely expensive to acquire in real-world, which often need specialized teleoperation systems with additional sensors and fine-grained calibration with high human laborer cost (Zhao et al., 2023a; Fu et al., 2024; Ding et al., 2024; Cheng et al., 2024; Wang et al., 2024; Wu et al., 2024). To address this challenge, recent methods aim to simplify learning budget by exploiting human inductive bias like parameterized atomic movements

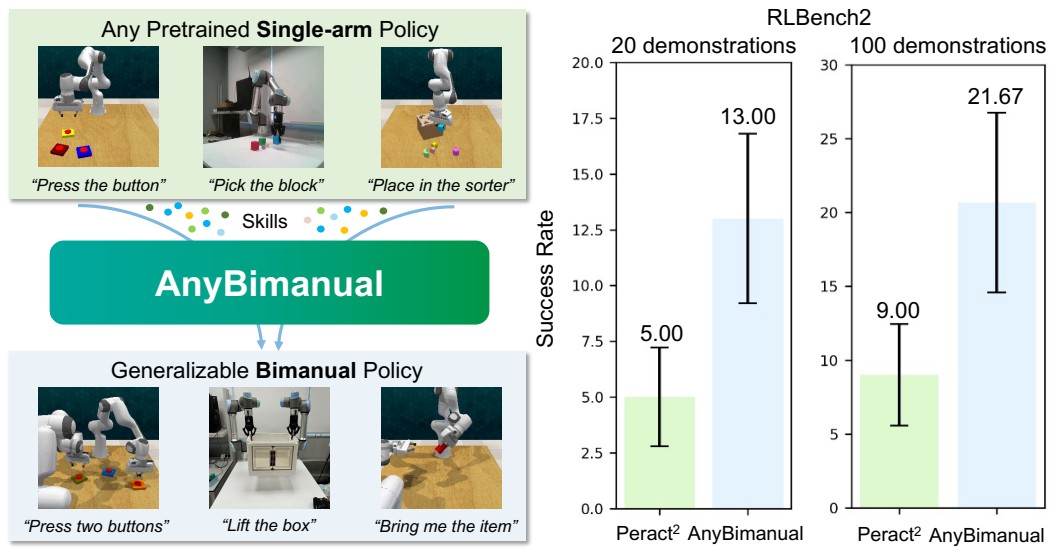

Figure 1: Left: the proposed AnyBimanual enables transferring from pretrained single-arm policies to bimanual manipulation policy, which preserves the generalizability with the proposed skill scheduling framework. Right: AnyBimanual surpasses the state-of-the-art multi-task bimanual manipulation agent (Grotz et al., 2024) by a large improvement of 12.67%.

(Chitnis et al., 2020b; Gong et al., 2023) or assigning stablizing and acting roles for each arm (Liu et al., 2024; Varley et al., 2024; Gao et al., 2024a), thereby reducing the need for extensive expert data. Nevertheless, shareable atomic movements and cooperation patterns across different bimanual manipulation tasks are often hard to specified even by human users, which limits the deployment scenario of these classes of methods.

In this paper, we propose a plug-and-play module named AnyBimanual that transfers any pre-trained single-arm policy to bimanual manipulation policy with limited demonstrations. Since single-arm policy model (Shridhar et al., 2022; Ke et al., 2024) has demonstrated impressive generalization ability across tasks due to the large model sizes and numerous training demonstrations, we realize high generalizability across diverse language-conditioned bimanual manipulation tasks by mining and transfering the commonsense knowledge in pretrained single-arm policies. More specifically, we first introduce a skill manager that dynamically schedules discovered skill primitives from the pretrained single-arm policy for language embedding boosting. Skill primitives demonstrates the shareable manipulation knowledge across embodiments, which are combined with importance weights and embodiment-specific compensation. To enhance the transderability of the pretrained single-arm policy in bimanual manipulation tasks, the observation discrepancy between single-arm and dual-arm systems should be minimized. We propose a voxel editor to generate spatial soft masks with for visual embeddings of the workspace, whose goal is to align visual input of single-arm pol-icy model for each arm with those during pretraining stage. Figure 3 shows an example of dual-arm manipulation policy composition from two single-arm policy models for the left and right arms. We evaluate AnyBimanual on a comprehensive task suite composed of 13 simluated tasks from RLBench2 (Grotz et al., 2024) and real-world tasks, where our method surpasses the previous state-of-the-art method by a large margin. The contributions are summarized as follows:

- We propose a model-agnostic plug-and-play framework named AnyBimanual that transfers an arbitrary pretrained single-arm policy to generalizable bimanual manipulation policy with limited bimanual demonstrations.

- We introduce a skill manager to dynamically schedule skill primitives for single-arm policy transferring and a voxel editor to mitigate the observation discrepancy between single-arm and dual-arm systems for transferability enhancement.

- We conduct extensive experiments of 13 tasks from RLBench2 and real world, and the results demonstrate that our method achieves a higher success rate than the state-of-the-art methods.

## 2 RELATED WORK

**Generalizable Bimanual Manipulation.** Generalizable Bimanual manipulation agent is able to handle a large variety of tasks by predicting a trajectory of dual-arm operation, which is with significant potential in complex applications from household service (Zhang et al., 2024), robotic surgery (Hu et al., 2023), to component assembly in factories (Buhl et al., 2019). To achieve multi-task learning, earlier studies attempted to leverage the emerged general understanding and reasoning capacities of pretrained foundation models like LLMs (Touvron & et al., 2023) and VLMs (Chen & et al., 2023), where the foundation model was prompted to generate high-level plan for low-level executors. For example, VoxPoser (Huang et al., 2023) utilized LLMs and VLMs to specify 3D affordance and constraint map via code generation, where a sampling-based motion planner is employed as the low-level executor. However, the performance of directly leveraging foundation models in a training-free manner is bottlenecked by the predefined low-level executor, which struggles to generalize to more contact-rich tasks like straightening a rope that are highly-desired in real-world applications. To overcome this challenge, robotic foundation models (Kim et al., 2024) that pretrained on large-scale real-world demonstrations were proposed under the single-arm setting, which have shown high generalizabilty across everyday manipulation tasks. However, bimanual tasks demand precise coordination of two high degree-of-freedom arms, making the teleoperation of demonstrations for training generalizable policies also costly. Although some recent approaches (Zhao et al., 2023a; Fu et al., 2024; Ding et al., 2024; Chuang et al., 2024; Yang et al., 2024) have developed more specialized teleoperation systems to reduce these costs, scaling up demonstrations for high generalization ability remains a challenge. To address the limited availability of demonstrations, alternative methods (Wang et al., 2021; Liu et al., 2024; Grannen et al., 2023) proposed to simplify the learning of bimanual policies by decoupling the dual-arm system into a stabilizing arm and an acting arm. Nevertheless, these methods often rely on predefined roles for each arm, which precludes their applicability to tasks requiring more flexible collaboration patterns. In contrast to these approaches, our work presents a novel method that transfers generalizable single-arm policies to bimanual tasks, which eliminates the necessity for explicit inductive bias like role specification.

**Skill-based Methods.** Skill learning (Zhang et al., 2023) is the process where intelligent agents acquire new abilities that are transferable across different tasks, which is of great significance for cross-task generalization. Thus, skill learning is being attractive in enhancing the generalizability of different models, such as game agents (Wang et al., 2023), robotic manipulation (Liang et al., 2024), and autonomous driving (Fu et al., 2023). The initial attempt to utilize skill learning was orchestrating a set of predefined skill (Munawar et al., 2018), which hindered their scalability to unseen tasks. To overcome this limitation, (Liang et al., 2024; Chitnis et al., 2020b) proposed to learn shareable skill primitives from data. For example, skill diffuser (Liang et al., 2024) introduced a hierarchical planning framework that integrating learnable skill embedding into conditional trajectory generation, which realized accurate execution of diverse compositional tasks. In the field of bimanual manipulation, skill learning was dominated by handcrafted primitives. For example, (Batinica et al., 2017; Colomé & Torras, 2020; Franzese et al., 2022; Grannen et al., 2020; Avigal et al., 2022; Ganapathi et al., 2021; Amadio et al., 2019; Yin & Chen, 2014; Fu et al., 2022; Chitnis et al., 2020a; Xie et al., 2020; Ha & Song, 2021) proposed to utilize parameterized atomic movements to shrink the high dimensionality of the bimanual action space, which has shown impressive performance on templated bimanual manipulation tasks. While the predefined atomic movements did boost the success rate on specific tasks, they are often difficult for even human users to specify, which restricts the deployment scenarios of these methods. In this paper, we propose to leveraging learnable implicit skill primitives to represent the learned commonsense of the pretrained single-arm policy, so that the knowledge can be transferred across different levels of tasks.

## 3 APPROACH

In this section, we first briefly introduce preliminaries on the problem formulation (Section 3.1), and then we present an overview of our pipeline (Section 3.2). Subsequently, we introduce a skill man-

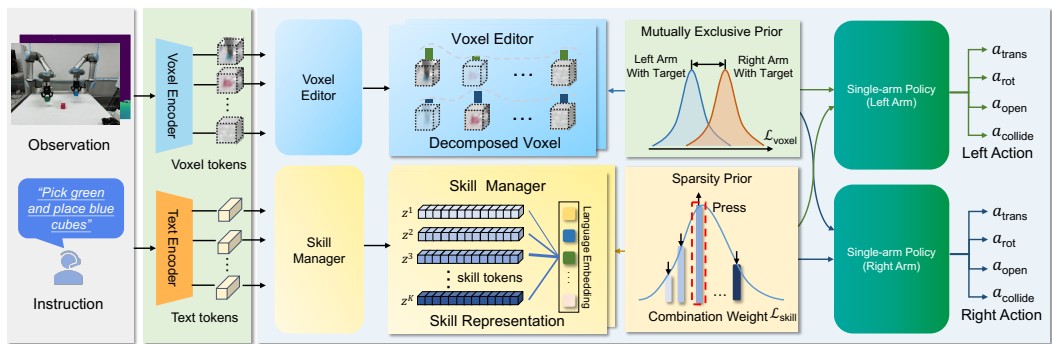

Figure 2: The overall pipeline of AnyBimanual, which primarily consists of a skill manager and a perception manager. The skill manager adaptively coordinates primitive skills for each robot arm, while the perception manager mitigates the distributional shift from single-arm to dual-arm by decomposing the 3D voxel input for each arm.

ager (Section 3.3) that schedules skill primitives to each arm to form the generalization bimanual manipulation policies. Finally, we build a voxel editor (Section 3.4) to mitigate the observation discrepancy between dual-arm and single-arm systems for policy generalization ability enhancement.

### 3.1 PROBLEM FORMULATION

The task of policy learning for bimanual manipulation can be defined as follows. To complete a wide range of manipulation tasks specified in natural language, the dual-arm agent is required to interactively predict actions of both end-effectors based on the visual observation and robot states, where the motion is acquired by a low-level motion planner (e.g., RRT-Connect). The observation $o_t$ at the $t_{th}$ time step includes the voxels $v_t$ converted from RGB and depth images (Grotz et al., 2024; Shridhar et al., 2022) and the robot proprioception $p_t$. The action $a_t$ for each end-effector at the $t_{th}$ time step contains the location $a_{\text{trans}}$, orientation $a_{\text{rot}}$, gripper open state $a_{\text{open}}$ and usage of collision avoidance in the motion planner $a_{\text{collide}}$ for goal reach. For the training data, human demonstrators produce $K$ offline expert trajectories $\mathcal{D} = \{(o_1, a_1^{\text{left}}, a_1^{\text{right}}), ..., (o_K, a_K^{\text{left}}, a_K^{\text{right}})\}$ for each task instruction, where $a_t^{\text{left}}$ and $a_t^{\text{right}}$ respectively demonstrate the actions for left and right grippers. Each expert trajectory also pairs with a natural language instruction $l$ that specifies the task goal. Existing methods tend to directly learn the policy model from expert demonstrations, which have shown effectiveness on single-task settings. However, due to the extremely high cost of data collection in dual-arm systems, the lack of expert demonstrations limits the generalizability of these methods across tasks. To address this, we present to transfer generalizable single-arm policy from pre-trained models for generalizable multi-task bimanual manipulation policies.

### 3.2 OVERALL PIPELINE

The overall pipeline of our AnyBimanual method is shown in Figure 2. For the language branch, we employed a pretrained text encoder (Radford et al., 2021) to parse the bimanual instruction to language embeddings with high-level semantics, where the skill manager schedules the skill primitives with composition and compensation to boost the language embeddings. Therefore, the pre-trained single-arm policy model can be prompted to generate feasible manipulation policy for each arm with high generalization ability across tasks with the sharable manipulation knowledge. For the visual branch, we lift and voxelize the RGB-D input to the voxel space as visual observation, and a 3D sparse voxel encoder is utilized to encode and tokenize the voxel observation for informative volumetric representation. The voxel editor generates a soft spatial mask to align visual representation of single-arm policy model with that during pretraining, so that the observation discrepancy between single-arm and dual-arm systems can be minimized for policy transferability enhancement. Finally, we employ two pretrained single-arm models to predict the left and right robot actions based on the visual representations and text embeddings, where the pretrained single-arm policy can be multi-modal transformer-based policy (Shridhar et al., 2022; Kim et al., 2024) or diffusion-based policy (Ke et al., 2024; Team et al., 2024).

### 3.3 DISCOVERING AND SCHEDULING TRANSFERABLE SINGLE-ARM SKILL PRIMITIVES

In order to transfer single-arm manipulation policy to dual-arm setting without generalizability drops, we propose a skill manager to decompose the action policies from single-arm foundation models into skill primitives and integrates primitives for dual-arm systems. However, the given offline expert demonstrations $\mathcal{D}$ does not contain any explicit intermediate skill primitives or sub-task boundaries, but only low-level end-effector poses and a high-level natural language instruction are provided. Therefore, we design a automatic skill discovery method in an unsupervised manner to learn primitive skills and their schema from offline bimanual manipulation datasets during training. In the test phase, the skill manager predicts different weighted combinations of discovered primitive skills to orchestrate each arm given high-level language instruction, which enables effective transfer of pre-trained single-arm policy to diverse bimanual manipulation tasks.

**Skill Manager.** We start with a discrete primitive skill set $\mathcal{Z} = \{z^1, z^2, ..., z^K\}$, where $K$ is a hyper-parameter that indicates the number of skill primitives. The trade-off between the expression ability of skill subspaces and the sparsity of the skill representation is considered by tuning the optimal $K$. To realize end-to-end skill discovery and scheduling, each potential skill is an implicit embedding $z^k \in \mathbb{R}^D$, which can be initialized with the corresponding language template tokens of the pretrained single-arm policy to mitigate the domain gap. By combining the primitives from the skill set, the language embedding for the single-arm policy model can be represented as a linear combination of the skill set. The reconstructed language embedding can be expressed as:

$$\hat{l}_t^{\text{left}} = \sum_{k=1}^{K} \hat{w}_{k,t}^{\text{left}} z_k + \epsilon_t^{\text{left}}, \quad \hat{l}_t^{\text{right}} = \sum_{k=1}^{K} \hat{w}_{k,t}^{\text{right}} z_k + \epsilon_t^{\text{right}} \quad (1)$$

where $\hat{l}_t^{\text{arm}}$ is the decomposed single-arm language embedding for one arm of the bimanual system, and $\hat{w}_t^{\text{arm}} \in \mathbb{R}^K$ denotes the importance weight for linear combination for both values of arm $\in \{\text{left, right}\}$. We also introduce a residual $\epsilon_t^{\text{arm}} \in \mathbb{R}^D$ to introduce the embodiment-specific knowledge for the policy transfer. The upscript arm of the variables can be selected from left or right to show the embodiment in the dual-arm systems. As depicted in Figure 3, considering a bimanual task Handover, it can be explicitly solved by scheduling two single-arm primitive skills, i.e., the left arm Place the block to the right gripper while the right arm Pick it from the left gripper.

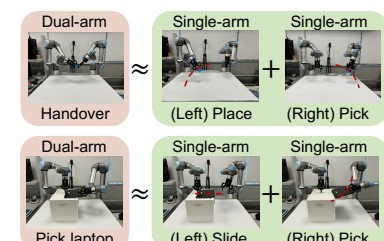

Figure 3: **Shareable skills across single-arm and dual-arm settings.** We observe that bimanual tasks are often originated from the combination of single-arm sub-tasks, which thus can be solved by effectively coordinating single-arm skills.

Though every language embedding that passed through the pretrained single policy can be represented as a linear combination of the skill set, the combination weight is unknown. We propose to parametrize a multi-model transformer named skill manager to dynamically predict the combination weight for each arm at each time step. Therefore, our skill manager can be formulated as $(\hat{w}_t^{\text{left}}, \epsilon_t^{\text{left}}, \hat{w}_t^{\text{right}}, \epsilon_t^{\text{right}}) = f(v_t, l, p_t)$, which takes the overall bimanual visual and language embeddings, proprioception as input, and assigns the reconstructed single-arm language embedding for each arm as output to schedule the skill primitive of each arm dynamically. Finally, the reconstructed single-arm language embedding is concatenated with the initial bimanual language embedding to enhance the global context, which is then forwarded to the corresponding single-arm policy.

**Learning Generalizable Skill Primitives.** To update the skill library, we expect the discoverd skill primitives are informative to encode fundamental robot motion that are sharable across a variety of tasks, thereby enhancing the generalizability of our framework. To realize this, the learning objective of the skill manager is designed as a sparse representation problem (Wright et al., 2010):

$$\mathcal{L}_{\text{skill}} = \|\hat{w}^{\text{left}}\|_1 + \|\hat{w}^{\text{right}}\|_1 + \lambda_\epsilon (\|\epsilon^{\text{left}}\|_{2,1} + \|\epsilon^{\text{right}}\|_{2,1}) \quad (2)$$

where $\| \cdot \|_1$, and $\| \cdot \|_{2,1}$ denote the $\ell_1$ and $\ell_{2,1}$ norm, respectively. $\lambda_\epsilon$ is a hyper-parameter that balances the denoising term. By minimizing this sparse regularization term, the skill manager is encouraged to reconstruct the language embedding by selecting less skills, which is further surrgated by minimizing a differentiable $\ell_1$ regularization (Tibshirani, 1996). Therefore, the skill subspaces is required to be orthogonal and disjoint with each other to reconstruct the language embedding with

sparse combination and compensation, which implicitly enforces each primitive skill to capture an independent fundamental motion.

### 3.4 DECOMPOSING VOLUMETRIC OBSERVATION WITH SPATIAL SOFT MASK

**Voxel Editor.** Despite the skill manager enables generalization in the language modality, the distributional shift from the single-arm to dual-arm workspace in terms of the visual context still may harm the model performance. To mitigate the observation discrepancy, we present a voxel editor $q$ that predict two spatial soft masks at each step $t$ to edit the voxel space so that the decomposed subspace of single-arm policy model for each arm aligns with those during pretraining stage: $(\hat{v}_t^{\text{left}}, \hat{v}_t^{\text{right}}) = q(v_t, l, p_t)$. The decomposed observation represents the locality of the workspace, which is then augmented by the bimanual observation to form the final visual embedding:

$$v_t^{\text{left}} = (\hat{v}_t^{\text{left}} \odot v_t) \oplus v_t, \quad v_t^{\text{right}} = (\hat{v}_t^{\text{right}} \odot v_t) \oplus v_t, \quad (3)$$

where $\odot$ is the element-wise multiplication, and $\oplus$ refers to the concatenation operator. As a result, the augmented visual representations for each arm contains both embodiment-specific information and the global context, which is then passed through the two single-arm policies to decode the optimal bimanual action.

**Learning Aligned Spatial Representation.** Our goal is to mitigate the visual domain gap between the single-arm and dual-arm setting, so that the pretrained commonsense knowledge in single-arm policy can be transferred with high adaptation ability. Since we can not access the single-arm pretraining data in common usages, we instead impose a mutually exclusive prior to the voxel editor that encourages it to implicitly divide the dual-arm workspace into single-arm workspaces. This prior is regularized by optimizing a Jensen-Shannon (JS) divergence regularization term:

$$\mathcal{L}_{\text{voxel}} = -D_{KL}(\hat{v}_t^{\text{left}} \| \hat{v}_t^{\text{right}})/2 - D_{KL}(\hat{v}_t^{\text{right}} \| \hat{v}_t^{\text{left}})/2 \quad (4)$$

where $D_{KL}$ means the Kullback-Leibler (KL)-divergence operator. To provide further explaination, bimanual manipulation tasks often involves asynchronous collaboration that requires the left and right arm attends to different areas of the whole workspace to act as different roles, such as stabilizing and acting. Based on this observation, the mutually exclusive division of the entire bimaual workspace will naturally separates one arm and its target from the other, which highly resembles the single-arm configuration. Hence by maximizing the divergence between the two soft masks, the voxel input of the bimanual manipulation agent can be disentangled into single-arm visual representations that align with those in the pretraining phase effectively.

### 3.5 LEARNING OBJECTIVES

The decomposed skill and volumetric representation are used to pass through the two pretrained single-arm policies to predict the optimal actions of the two end-effectors. We assume access to a pretrained single-arm policy $p$, which is fundamentally a multi-model multi-task neural network that takes visual and language embedding as inputs and outputs 6 DoF robot arm. Our AnyBimanual is a model-agnostic plug-and-play method, which indicates that the architecture of pretrained single-arm policy $p$ is flexible in different architectures such as multi-modal transformer-based policies (Shridhar et al., 2022; Kim et al., 2024) and diffusion policies (Ke et al., 2024; Team et al., 2024). To supervise the model with the provided expert demonstrations for behavior cloning, we leverage the cross-entropy loss to learn accurate action prediction for each arm:

$$\mathcal{L}_{\text{action}} = CE(p_{\text{trans}}^{\text{left}}, p_{\text{rot}}^{\text{left}}, p_{\text{open}}^{\text{left}}, p_{\text{collide}}^{\text{left}}) + CE(p_{\text{trans}}^{\text{right}}, p_{\text{rot}}^{\text{right}}, p_{\text{open}}^{\text{right}}, p_{\text{collide}}^{\text{right}}) \quad (5)$$

where $p_{\text{trans}}^{\text{arm}}, p_{\text{rot}}^{\text{arm}}, p_{\text{open}}^{\text{arm}}, p_{\text{collide}}^{\text{arm}}$ represents the distribution of the ground-truth actions for translation, rotation, gripper openness, and collision avoidance for the corresponding robot arm, respectively. The behavior cloning loss is then combined with the two regularization terms described above to learn informative skill manager and voxel editor. To conclude, the overall training objective of AnyBimanual is:

$$\mathcal{L}_{\text{total}} = \mathcal{L}_{\text{action}} + \lambda_{\text{skill}} \mathcal{L}_{\text{skill}} + \lambda_{\text{voxel}} \mathcal{L}_{\text{voxel}}, \quad (6)$$

where $\lambda_{\text{skill}}$ and $\lambda_{\text{voxel}}$ refer to hyper-parameters that balance the importance of the regularizations.

## 4 EXPERIMENTS

In this section, we first introduce the experimental setup including datasets, baseline methods and implementation details (Section 4.1). Then, we compare our method with the state-of-the-art approaches to show its superiority (Section 4.2), and conduct a comprehensive ablation study to evaluate the validity of various components in our AnyBimanual framework (Section 4.3). Finally, we present qualitative results to depict the effectiveness of our AnyBimanual in real-world settings, and interpret the learned skill primitives and decomposed volumetric representation by visualization (Section 4.4). More results and case studies can be found in the appendix.

### 4.1 EXPERIMENT SETUP

**Simulation.** For benchmarking, our simulation experiments are conducted on RLBench2 (Grotz et al., 2024), a bimanual version extended from the widely-used RLBench (James et al., 2019) benchmark in prior works (Jaegle et al., 2021; Ze et al., 2024; Lu et al., 2024; Goyal et al., 2023; Xian et al., 2023; Ke et al., 2024). Following the setup in (Grotz et al., 2024), we utilize a subset of 12 languaged-conditioned manipulation tasks varing from different challenge levels. The diversity of this task suite requires the agent to acquire shareable skill primitives and schedule them correctly according to the natural language to achieve high success rates, rather than simply imitating limited expert demonstrations. For agent observation, we employed six cameras (i.e. front, left, right, wrist left, wrist right, overhead) with a resolution of $256 \times 256$ to cover the entire workspace and two robot arms. For behavior cloning in the training phase, we provide 100 demonstrations for each task, which are generated by a oracle script expert. In the test phase, we evaluate 25 episodes per task in the testing set to mitigate bias from noise.

**Real Robot.** The real-world setup for our experiments involves two Universal Robots UR5e manipulators equipped with Robotiq 2F-85 grippers, controlled using two Xbox gamepads as 6-DoF controllers for collecting demonstrations. An RGB-D Realsense camera mounted on a tripod provides $640 \times 480$ resolution images at 30 Hz, simulating a human viewpoint during bimanual tasks. Camera-to-arm calibration is achieved using the easy_handeye package and ARUCO markers on the end-effectors. We collect 20 real-world human demonstrations during training, while the evaluations are conducted using a Nvidia RTX 4080 GPU. For more setup details, see Section 4.5.We present two qualitative examples of the action sequences in Figure 4.

**Baselines.** We compare our AnyBimanual with the state-for-the-art approaches, including PerAct[2] (Grotz et al., 2024), which is a strengthened version from the well-known single-arm policy PerAct. To exclude the influence of model parameter, we also implement a baseline that incorporates two pre-trained multi-task PerAct (Shridhar et al., 2022) model. Additionally, we include PerAct-LF in our comparisons, which employs a leader-follower Grotz et al. (2024) architecture using two Perceiver Actor networks. The evaluation metric is the task completion rate or success rate, which is defined as the percentage of episodes where the agent successfully completes the goal specified by natural language within a budget of 25 steps.

**Implementation Details.** Following the common training recipe (Shridhar et al., 2022; Zhao et al., 2023b), we use the SE(3) augmentation for the expert demonstrations in the training set to improve the generalizability of the agents. Specifically, we augment each training sample by perturbing the 3D point cloud with $[\pm0.125m, \pm0.125m, \pm0.125m]$ translation, and rotate it around the $z$-axis by $[0°, 0°, 45°]$. All compared methods are trained on two NVIDIA RTX 3090 GPUs for 100k iterations with a batch size of 2. We use the LAMB optimizer (You et al., 2020) with in a constant learning rate of $5 \times 10^{-4}$ to update the model parameters.

### 4.2 COMPARISON WITH THE STATE-OF-THE-ART METHODS

In this section, we compare our AnyBimanual with previous stat-of-the-art approaches on RLBench tasksuite. Table 1 presents a comparison of the average success rates for each task and the average performance is shown in Figure 1. Our method achieves the best overall performance, with an average success rate of 21.67%, outperforms the previous perception-based methods, setting a new state-of-the-art. The perception-based method PerAct[2], which builds on the PerAct framework, by utlizing a voxel-based representation that is robust to changes in viewpoint, which showed effective improvement beyond the methods that rely on rendered vitural images or joint angles like

Table 1: **Multi-Task Test Results.** Mean success rates (%) of multi-task agents trained with 20 or 100 demonstrations per task and evaluated over 25 episodes. The average performance is shown in Figure 1.

| Method | pick laptop 20 | 100 | pick plate 20 | 100 | straighten rope 20 | 100 | lift ball 20 | 100 | lift tray 20 | 100 | push box 20 | 100 |
|---|---|---|---|---|---|---|---|---|---|---|---|---|
| PerAct-LF | 0 | 0 | 0 | 4 | 4 | 8 | 4 | 12 | 0 | 4 | 8 | 16 |
| **PerAct-LF + AnyBimanual** | 0 | 4 | 0 | 4 | 4 | 4 | 8 | 12 | 0 | 8 | 12 | **24** |
| PerAct$^2$ | 0 | **8** | 4 | 16 | 0 | 4 | 8 | **40** | 0 | 4 | 0 | 0 |
| PerAct$^2$ + Pretraining | 0 | 4 | 0 | 20 | 4 | 12 | 16 | 12 | 0 | **12** | 0 | 8 |
| **PerAct$^2$ + AnyBimanual** | **8** | 8 | **12** | **32** | **8** | **24** | 4 | 32 | **4** | **12** | **28** | 16 |

| Method | put in fridge 20 | 100 | press buttons 20 | 100 | handover item 20 | 100 | sweep to dustpan 20 | 100 | take out tray 20 | 100 | handover easy 20 | 100 |
|---|---|---|---|---|---|---|---|---|---|---|---|---|
| PerAct-LF | 0 | 0 | 0 | 8 | 0 | 0 | 0 | 0 | 0 | 0 | 0 | 4 |
| **PerAct-LF + AnyBimanual** | 0 | 4 | 4 | 8 | 0 | 0 | 0 | 0 | 0 | 0 | 0 | 8 |
| PerAct$^2$ | 0 | 0 | 4 | 12 | **8** | 0 | **8** | 0 | 0 | 0 | 28 | 24 |
| PerAct$^2$ + Pretraining | 0 | 0 | 44 | 64 | 0 | 4 | 0 | 8 | 8 | 4 | 16 | **28** |
| **PerAct$^2$ + AnyBimanual** | 0 | **8** | **48** | **84** | **8** | **8** | 4 | 4 | **8** | **4** | 24 | **28** |

RVT Goyal et al. (2023). Despite its novel dual-arm architecture, PerAct$^2$ does not fully exploit the generalization capabilities demonstrated by single-arm models, such as PerAct, which have been highly effective across various manipulation tasks. The lack of knowledge transfer from these effective single-arm policies limits the performance of PerAct$^2$ in complex bimanual tasks. In contrast, our AnyBimanual method leverages the knowledge distilled from single-arm models and successfully transfers it to guide dual-arm manipulation. This strategic integration enables more precise and context-aware action prediction. As a result, our method outperforms PerAct$^2$ by a significant improvement of 12.67% on average.

We further observe that our AnyBimanual exhibits a more significant improvement in long-horizon, multi-stage tasks compared to simpler, short-horizon tasks. For instance, in tasks such as handover and oven, which require continuous coordination between both arms over a longer period, our method shows a more pronounced performance boost compared to 0% success rate of PerAct$^2$. This can be attributed to AnyBimanual 's ability to dynamically manage and adapt skill combinations over time, which is particularly crucial in tasks with evolving scene dynamics. In contrast, for shorter, single-action tasks like pressing buttons, the performance improvement, while still present, is less dramatic, as these tasks rely more on straightforward manipulation. This phenomenon suggests that AnyBimanual is especially effective in complex, longer-horizon tasks where adaptive action prediction and coordination are critical. It reinforces the idea that transferring knowledge from single-arm models, combined with the skill managing mechanism, allows for more flexible and precise action generation in diverse and challenging environments.

## 4.3 ABLATION STUDY

Our AnyBimanual framework leverages the combination of a skill manager and a voxel editor to dynamically coordinate single-arm skills while mitigating the distributional shift between single-arm and dual-arm visual inputs. We conduct an ablation study, as shown in Table 2, to validate the effectiveness of each component. First, we implement a vanilla baseline without any of the proposed techniques, where the model loads the pre-trained PerAct model and directly train the perception-based model to predict robot action.

**Skill Manager.** By adding a one-hot selection mechanism to select a single skill from the skill set during training, the performance improves by 2% compared to the baseline. We then replace the one-hot selection mechanism with a linear combination of all skills from the skill set, rather than relying on a single skill. By employing the linear combination approach, along with the associated residual, we observe that the average success rate increases by 2.67% compared to using a single skill, which indicates the enhanced generalization ability of the model across tasks in robotic manipulation. Especially, in the tasks that require effective skill managing, such as Long, Planning, Motion and Occlusion, it outperforms the vanilla version by sizable margins, which demonstrates the skill manager's effectiveness in managing complex interactions, handling high variability, and adapting to long-horizon tasks within dynamic environments.

**Voxel Editor.** Additionally, we incorporate the spatial soft masking from the voxel editor, resulting

Table 2: **Comparison of Our Methods with Different Techniques.** We manually categorize the 12 RL-Bench2 task to 6 groups for further interpretability. For more details, please refer to the appendix.

| One-hot Skill. | Skill Manager. | Voxel Editor. | Long | Planning | Tools | Motion | Lift | Occlusion | Average |
|:---:|:---:|:---:|:---:|:---:|:---:|:---:|:---:|:---:|:---:|
| ✗ | ✗ | ✗ | 12 | 64 | _8_ | 12 | 12 | 0 | 14.33 |
| ✓ | ✗ | ✗ | _16_ | 72 | 4 | 20 | 12 | **8** | 16.33 |
| ✗ | ✓ | ✗ | **18** | _76_ | 4 | **36** | 15 | _4_ | _19.00_ |
| ✓ | ✗ | ✓ | 14 | **84** | 6 | 20 | _16_ | 8 | 18.67 |
| ✗ | ✓ | ✓ | 18 | 84 | 10 | _24_ | **17** | 8 | **21.67** |

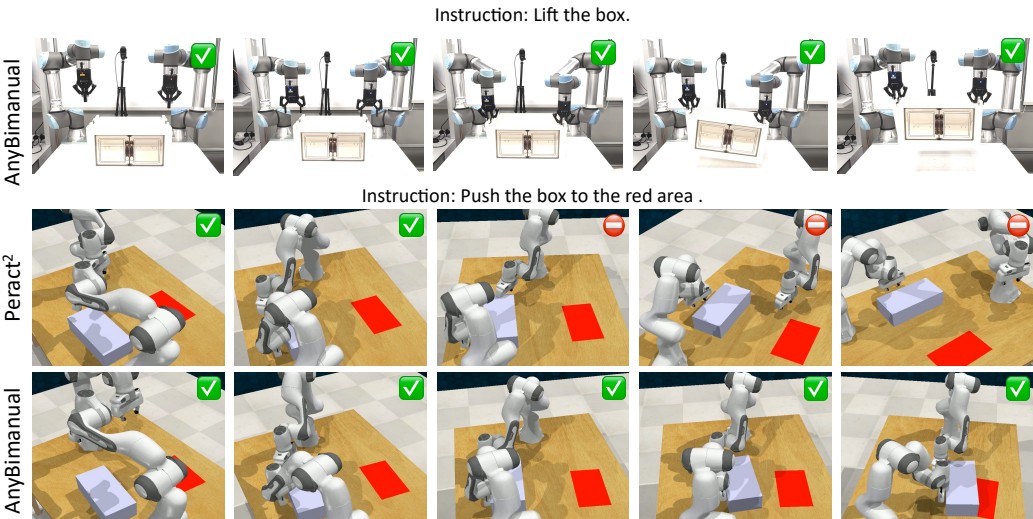

Figure 4: **Examples of Our AnyBimanual in the Real World.** AnyBimanual can conduct complex real-world bimanual manipulation tasks simultaneously with only 20 human demonstrations per task.

in a substantial performance improvement of 2.67%. Although the inclusion of spatial soft masking mechanism may slightly affect performance in simpler tasks due to the added complexity of input processing, it leads to significant gains in overall task success. Notably, in `Tools` and `Hoist` tasks, which demand a high degree of highly coordination and task differentiation between the left and right arms, the voxel editor shows remarkable effectiveness. This demonstrates its ability to improve spatial decomposition and action precision in tasks where each arm must perform distinct yet coordinated actions. By integrating all components in our AnyBimanual framework, the success rate improves from 14.33% to 21.67%, validating the importance of leveraging single-arm knowledge and 3D scene soft mask to achieve superior performance in bimanual robotic manipulation.

### 4.4 QUALITATIVE ANALYSIS

We present two qualitative examples of the generated action sequences in Figure 4, comparing PerAct[2] and our AnyBimanual method. In the top case, the agent is instructed to "Lift the box." The right arm first grasps the box, followed by the left arm securing its hold. Both arms then lift the box simultaneously, demonstrating precise and coordinated bimanual manipulation. In the bottom case, the instruction is "Push the box to the red area". The PerAct[2] agent fails to touch the box and only mimics the expert's forward pushing motion, resulting in an incomplete action. In contrast, our AnyBimanual agent ensures both arms make contact with the box and successfully pushes it into the red area, demonstrates our method's superior ability in having each arm correctly identify contact with the object and successfully complete the task.

We also visualize the linear combination of the skill set at different timesteps and the decomposition of volumetric observation to further illustrate the effectiveness of our AnyBimanual method, which is shown in Figure 5. In the skill manager, we use 18 task embeddings from PerAct (Shridhar et al., 2022) as skill set. At timestep 1, the right arm primarily follows the "place blocks in sorter" task ($skill$ 7) from the skill set, as the right arm needs to pick up a block from the table at first, similar to

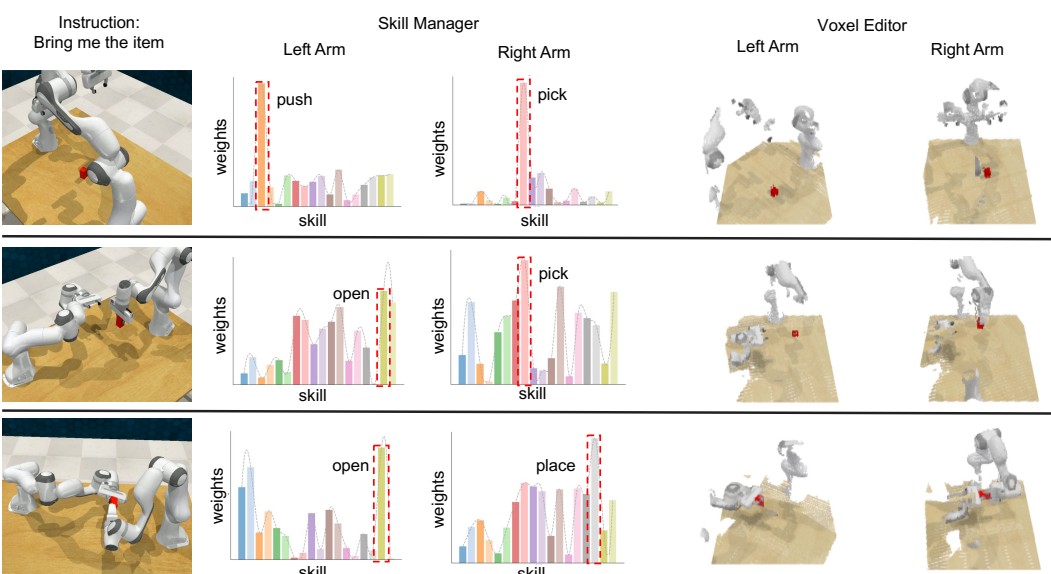

Figure 5: **Visualization of Our AnyBimanual.** This figure shows in different key timesteps, how the skill manager dynamically adjusts skill weights for each arm and how the voxel editor decomposes volumetric observation.

the sorting task, meanwhile, the left arm is guided by the "push button" task ($skill$ 2), as its motion only involves a downward push without interacting with any objects, very similar to the push button scenario. By timestep 5, after the right arm has picked the red block, it remains stationary while the left arm approached and grasps the block, similar to the "open drawer" task ($skill$ 16) where the robotic arm approaches and grasps the drawer handle. Additionally, we can observe that the voxel editor effectively decomposes the voxel space, enabling each arm to mainly focus on information revelant to its own actions. For instance, in the voxel outputs for the left arm, some information from the right arm, such as the gripper, is soft-masked. This facilitates better spatial awareness and task coordination, enhancing the overall effectiveness of bimanual manipulation.

### 4.5 REAL-ROBOT RESULTS

We further validated our approach through read-robot experiments on two Universal Robots UR5e. For setup details, refer to Appendix A.2. Without any sim-to-real transfer, we trained a multi-task AnyBimanual agent from scratch on the `Lift` tasks (with 3 unique variations), which utilizes 20 demonstrations for training. Consistent with the simulation results, AnyBimanual achieved over 66% success on short-horizon tasks like lifting the box, while the state-of-the-art method PerAct[2] fails to learn effective policy given limited demonstrations.

### 5 CONCLUSION

In this paper, we have introduced AnyBimanual, a novel framework designed to transfer pretrained single-arm manipulation policies to multi-task bimanual manipulation with limited dual-arm demonstrations. We develop a skill manager to dynamically schedule skill primitives discovered from single-arm policies, enabling their effective adaptation for bimanual tasks with the addition of embodiment-specific compensations. To address the observation discrepancies between single-arm and bimanual systems, we propose a voxel editor that generates spatial soft masks, aligning the visual embeddings of each arm with those used during the pretraining stage of the single-arm policy model. Extensive experiments across 13 simulated and real-world tasks demonstrate the superiority of AnyBimanual. The limitations of our approach primarily arise from the need for careful visual alignment between single-arm and dual-arm systems to ensure robust policy transfer across tasks.

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

# A ADDITIONAL EXPERIMENTAL DETAILS

## A.1 SIMULATION

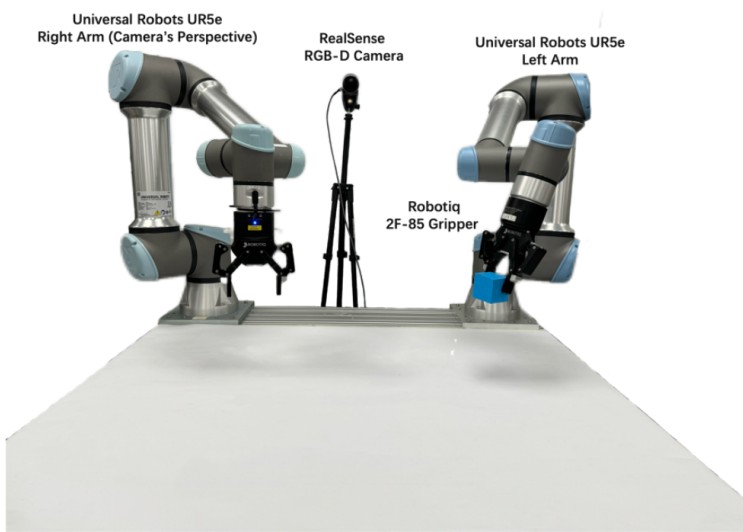

Figure 6: **Real-Robot Setup** with RealSense RGB-D Camera and Two UR5e Manipulators.

We utilize RLBench2 Grotz et al. (2024) as our main task suite for mulit-task learning. Table 4 is an overview of the 12 selected tasks we use in the experiments. Table 3 is an overview of the 18 selected tasks used to pretrain the single-arm checkpoint. Our task variations include randomly sampled colors, sizes, counts, placements, and categories of objects. Other properties vary depending on the specific task. Furthermore, objects are randomly arranged on the tabletop within a certain range, adding to the diversity of the tasks. In the ablation study, we adopt the task classification from Guhur et al. (2023) to group the RLBench2 tasks of Table 4 into 6 categories according to their key challenges. The task groups include:

- The `Planning` group contains tasks with multiple subtasks. The included tasks are: `dual push buttons`.
- The `Long` group includes long-term tasks that requires more than 7.5 keyframes. The included tasks are: `handover item` and `handover item easy`.
- The `Tools` group requires the agent to grasp an object to interact with the target object. The included tasks are: `coordinated push box` and `bimanual sweep to dustpan`.
- The `Motion` group requires precise control, which often causes failures due to the predefined motion planner. The included task is: `bimanual straighten rope`.
- The `Hoist` group requires both arms to lift the object to a certain height. The included task is: `pick laptop`, `pick plate`, `lift ball` and `lift tray`.
- The `Occlusion` group involves tasks with severe occlusion problems from certain views. The included task is: `put bottle in fridge` and `put bottle in fridge`.

## A.2 REAL-ROBOT

**Hardware Setup.** The real robot setup uses two Universal Robots UR5e manipulators, each equipped with a Robotiq 2F-85 gripper. See Figure 6 for reference. For the perception, we use a Realsense RGB-D camera mounted on a tripod, positioned to mimic the viewpoint of human eyes during bimanual tasks. The Realsense provides RGB-D images at a resolution of 640x480 with a frame rate of 30 Hz. The extrinsics between the camera and right arm base-frame are

Table 3: **Single-arm Tasks.** This table shows the 18 single-arm tasks in RLBench that used to pretrain the single-arm policy.

| Task | # of Variations | # of Average Keyframes | Human Instruction Template |
|---|---|---|---|
| open drawer | 3 | 3.0 | *"open the __ drawer"* |
| slide block | 4 | 4.7 | *"slide the block to __ target"* |
| sweep to dustpan | 2 | 4.6 | *"sweep dirt to the __ dustpan"* |
| meat off grill | 2 | 5.0 | *"take the __ off the grill"* |
| turn tap | 2 | 2.0 | *"turn __ tap"* |
| put in drawer | 3 | 12.0 | *"put the item in the __ drawer"* |
| close jar | 20 | 6.0 | *"close the __ jar"* |
| drag stick | 20 | 6.0 | *"use the stick to drag the cube onto the __ target"* |
| stack blocks | 60 | 14.6 | *"stack __ __ blocks"* |
| screw bulb | 20 | 7.0 | *"screw in the __ light bulb"* |
| put in safe | 3 | 5.0 | *"put the money away in the safe on the __ shelf"* |
| place wine | 3 | 5.0 | *"stack the wine bottle to the __ of the rack"* |
| put in cupboard | 9 | 5.0 | *"put the __ in the cupboard"* |
| sort shape | 5 | 5.0 | *"put the __ in the shape sorter"* |
| push buttons | 50 | 3.8 | *"push the __ button, [then the __ button]"* |
| insert peg | 20 | 5.0 | *"put the ring on the __ spoke"* |
| stack cups | 20 | 10.0 | *"stack the other cups on top of the __ cup"* |
| place cups | 3 | 11.5 | *"place __ cups on the cup holder"* |

Table 4: **Dual-arm Tasks.** This table shows the 12 dual-arm tasks in RLBench2 benchmark.

| Task | # of Variations | # of Average Keyframes | Human Instruction Template |
|---|---|---|---|
| push box | 1 | 2.1 | *"push the box to the red area."* |
| lift a ball | 1 | 4.0 | *"lift the ball."* |
| push two buttons | 5 | 4.0 | *"push the __ and __ button."* |
| pick up a plate | 1 | 6.6 | *"pick up the plate."* |
| put bottle in fridge | 1 | 7.8 | *"put the bottle into the fridge."* |
| handover an item | 5 | 7.6 | *"hand over the __ item."* |
| pick up notebook | 1 | 7.2 | *"pick up the notebook."* |
| straighten rope | 1 | 5.9 | *"straighten the rope."* |
| sweep dust pan | 1 | 7.3 | *"sweep the dust to the pan."* |
| lift tray | 1 | 5.1 | *"lift the tray."* |
| handover item (easy) | 1 | 7.5 | *"hand over the item."* |
| take tray out of oven | 1 | 8.7 | *"take tray out of oven."* |

calibrated using the easy_handeye package. Additionally, an ARUCO marker attached to the UR5e's end-effector is employed to aid in the calibration process.

**Data Collection.** We collect demonstrations with two Xbox gamepads. Each gamepad is a 6-DoF controller. The gamepads adjust the velocity of the arm's end-effector to translate and rotate in all directions, with reference to the arm's base-frame. For motion planning, we utilized the Universal Robots ROS Driver and MoveIt.

**Training and Execution.** We train an AnyBimanual agent using 20 demonstrations for each task, incorporating translational and rotational perturbations into the training samples to enhance the model's robustness. Each task undergoes training for a full day on a single Nvidia RTX 3090 GPU with a batch size of 2. During the evaluation phase, we select the final checkpoint form training, as there is no effective method available for assessing the model's performance during the training period. We perform inference on a single Nvidia RTX 4080 GPU.

