# OpenReview forum: "AnyBimanual: Transferring Single-arm Policy for General Bimanual Manipulation"
_ICLR.cc/2025/Conference — ICLR 2025 Conference Withdrawn Submission_

### Official Review · Reviewer_rJoW · 2024-10-28

**Soundness:** 3
**Presentation:** 3
**Contribution:** 2
**Rating:** 3
**Confidence:** 3

**Summary:**

This paper focuses on converting a single-arm manipulation policy into a dual-arm policy. A voxel editor is proposed to decompose bimanual observations into single-arm's observation, closing the visual gap, while a skill manager is proposed to combine primitive arm skills effectively, addressing the action gap. Both simulation and real-world experiments are conducted.

**Strengths:**

Learning bimanual manipulation is indeed a challenging task, as most robot datasets are single-arm. The approach of voxel editing and a skill manager to decompose the dual-arm policy into two single-arm policies, leveraging pretrained single-arm models, is interesting and may help reduce the data needed for dual-arm policy learning.

**Weaknesses:**

1. As emphasized in the paper, the main contribution is converting a single-arm policy into a bimanual policy. However, in the experiments, PerAct2 and PerAct-Lf already appear to be bimanual policies (e.g., the policy outputs left and right actions), which raises questions about how the proposed method is combined with such baselines. From my perspective, the experiments might be more insightful if baselines like PerAct were used instead. I expected to see that using a true single-arm policy with the proposed method could achieve comparable results with bimanual manipulation policies.

2. According to the original PerAct2 paper [1], one point of confusion is that the baseline success rates in this study seem much lower than those reported in the original paper. Am I missing some evaluation or condition differences?

3. One of the challenges in dual-arm versus single-arm manipulation is coordinating the two arms. The current method decomposes the dual-arm policy into two single-arm policies, seemingly without coordination between the arms; it seems that coordination is learned implicitly through imitation of expert demonstrations？ However, the supplementary demos provided seem to show significant asynchrony between the two arms' motions.

[1] Grotz, M., Shridhar, M., Asfour, T., & Fox, D. (2024). Peract2: A perceiver actor framework for bimanual manipulation tasks. arXiv preprint arXiv:2407.00278.

**Questions:**

1. The paper attempts to demonstrate that this method is plug-and-play. Conducting experiments on other pipelines, such as diffusion policy, rather than different versions of PerAct, would seem more reasonable.
2. The skill manager appears to require specific primitive skill descriptions in the instructions. What would happen if the instruction were simply "give me the Coke"?

---

### Official Review · Reviewer_X4vy · 2024-10-29

**Soundness:** 2
**Presentation:** 3
**Contribution:** 2
**Rating:** 3
**Confidence:** 4

**Summary:**

This work proposes a method for adapting pre-trained single-arm policies, with additional design choices, to enable generalization to bi-manual manipulation tasks. Specifically, given expert bi-manual manipulation data, the authors first identify skill prototypes from the demonstration data. They then use a skill manager to output linear combination weights for these prototypes, constructing a new skill embedding as input to the single-arm policy. To address observation discrepancies between each single arm, they introduce a voxel editor that outputs soft masks applied to the voxel space, ensuring that the decomposed subspaces of the single-arm policies align with those used during pre-training. Finally, an action reconstruction loss is employed to reproduce action labels in the demonstration data. Through various simulations and real-world experiments, the authors demonstrate the approach's effectiveness.

The main contributions of this paper are: (i) a novel approach toward generalizing single-arm policies to bi-manual manipulation tasks, (ii) a comprehensive and well-structured problem formulation, and (iii) clear and accessible writing throughout the paper.

**Strengths:**

The strengths of this paper are:

Originality: The integration of two single-arm policies to tackle bi-manual tasks presents a novel approach that adds value to the field. The authors have conducted an extensive literature review, covering a wide range of relevant areas that contextualize their work within the existing body of research. This thorough review not only highlights the gaps in current methodologies but also emphasizes the innovative aspects of the proposed framework, positioning it as a meaningful contribution to the study of bi-manual manipulation.

Quality: While the proposed method demonstrates potential, there are several critical details missing in the methodology section that hinder a full understanding of the approach. Additionally, the experiments presented are somewhat limited in scope. Expanding the experimental setup to include a wider variety of tasks (especially real-world experiments), metrics, and evaluations would strengthen the overall quality of the research. Please refer to the "Weaknesses" section for more detailed critiques regarding these issues.

Clarity: The writing throughout the paper is generally clear and accessible, making it easy for readers to follow the authors’ arguments and understand the proposed framework.

Significance: Since the reported performance of the proposed method is not particularly strong—despite being better than the baselines—it is challenging to ascertain the true significance of this work without further context. To enhance the paper's impact, it would be beneficial for the authors to provide additional insights into the specific challenges encountered in these tasks that may not be immediately intuitive. Understanding these complexities could help justify the results and potentially reshape the perception of the contributions made by this research.

**Weaknesses:**

There are several concerns:

1). In the problem formulation, the authors specify that they collect K offline expert trajectories. However, they later mention that expert demonstrations are "lacking" due to high data collection costs, which is confusing—since they still start with K trajectories. Clarifying this point would help improve coherence in the problem setup. Additionally, recent studies (e.g., [1], [2]) have developed low-cost tools for gathering data across various manipulation tasks. It would be helpful to understand the authors' perspective on whether easy data collection tools for single-arm tasks could also benefit bi-manual tasks, given that humans can intuitively control both arms in data collection.

[1]. Chi, Cheng, et al. "Universal manipulation interface: In-the-wild robot teaching without in-the-wild robots." arXiv preprint arXiv:2402.10329 (2024).

[2]. Yu, Kelin, et al. "MimicTouch: Leveraging Multi-modal Human Tactile Demonstrations for Contact-rich Manipulation." 8th Annual Conference on Robot Learning.

2). The authors use the collected K expert bi-manual trajectories to train the skill manager, which dynamically outputs importance weights for the linear combination of learnable skill primitives. However, how is the mapping established between the generated language embedding for one arm and its corresponding low-level control commands? Is this mapping solely managed by pre-trained models? If so, including some explanation of these details—even if they’re based on prior work—would greatly enhance the readability and understanding of this work.

3). In the learning objective for skill primitives, does this objective improve the low-level performance of the single-arm policy? Since the skill manager ultimately outputs a composite language embedding, how does the quality of this embedding impact the effectiveness of the bi-manual policy? There appear to be gaps in explaining how the skill manager’s outputs—used as inputs to the pre-trained low-level single-arm policy—connect to the policy’s low-level performance. To my understanding, learning the skill primitives does not enhance the single-arm policy’s performance directly. Instead, this approach seems focused on dynamically assigning high-level manipulation roles to each arm for a cooperative task rather than enhancing bi-manual manipulation precision. If this is the case, then the actual low-level manipulation performance is primarily driven by the reconstruction loss upon the K demonstrations, which isn’t a contribution unique to this work.

4). Further, the composite language embedding is concatenated with the original bi-manual language embedding. Could this final embedding be somewhat out-of-distribution (OOD) for the pre-trained single-arm policy model? For instance, commands like “push two buttons” would be an unseen task during single-arm policy training, so how would the pre-trained model handle such cases?

5). Similarly, the single-arm policies use augmented visual embeddings, but is the global context from bi-manual manipulation potentially out-of-distribution (OOD) for the frozen single-arm policy during pre-training? Even if Transformer-based single-arm policies can handle a variable number of input tokens, the global context tokens would still be OOD in this scenario. While the use of KL divergence to learn better decomposed subspaces makes sense to me, I’m unclear about the direct concatenation of the global context. If I’ve misunderstood this approach, could the authors provide clarification?

6). The performance of all policies, including the one proposed in this work, is notably low. This raises questions about whether imitation learning (IL) methods are well-suited for such tasks; reinforcement learning (RL) might potentially yield better results, or perhaps more data augmentation could improve raw IL data. I’d like to understand why the policy performs so poorly across most tasks—particularly for simpler tasks like lifting a ball, where success rates are only 32% despite training on 100 demonstrations. Are there additional challenges that are not immediately intuitive? If so, describing these in the paper would help clarify. Otherwise, the low success rate could indicate issues with the quality of the collected demonstrations or possible implementation bugs.

7). In Table 2, the average performance using no proposed methods achieves a success rate of 14.33%, while utilizing all proposed methods only improves this to 21.67%. This increase does not seem significant. Similar to my previous question regarding the poor overall performance, I would appreciate some insights into why the performance remains low despite the implementation of these methods. Understanding the factors contributing to this limited improvement would be beneficial.

8). In Figure 5, I’m puzzled by the "open" operations in the simple task "bring me the item." Specifically, it’s unclear why the left arm grasping the block is treated similarly to the open drawer task (line 513). Additionally, the visualization of the voxel editor is challenging to evaluate. Would it be possible to employ a better visualization tool, or is this approach common in the literature? Clarification on these points would enhance understanding.

9). The real-robot results presented in this paper are insufficient, as the authors only mention them toward the end. Additionally, the reported success rate of 66% for the lift task feels quite limited to fully demonstrate the effectiveness of the method. There is also a lack of evaluation using qualitative or other quantitative metrics to provide a comprehensive assessment of performance. Since this paper proposes a framework aimed at addressing challenging bi-manual tasks, including more extensive real-robot results and diverse metrics would be necessary to strengthen the contributions.

**Questions:**

See weakness section. If my concerns are decently addressed, I am happy to increase the rating.

---

### Official Review · Reviewer_BAU6 · 2024-11-02

**Soundness:** 3
**Presentation:** 2
**Contribution:** 3
**Rating:** 6
**Confidence:** 4

**Summary:**

In this paper, the authors introduce AnyBimanual, a framework for transferring pretrained single-arm manipulation policies to multi-task bimanual manipulation. They develop a skill manager to schedule skill primitives identified from single-arm policies for bimanual tasks, and propose a voxel editor to generate spatial soft masks that address the observational discrepancies between single-arm and bimanual systems. The authors conduct experiments across multiple tasks, comparing their approach against baselines and ablated versions.

**Strengths:**

* The idea of transferring pretrained single-arm policies to multi-task bimanual manipulation is both novel and appealing.
* Baseline comparisons and ablation studies demonstrate the impact of the proposed method.
* The qualitative visualizations enhance the understanding of the proposed approach and its results.

**Weaknesses:**

* To transfer the pretrained single-arm policies to multi-task bimanual manipulation, much of the contribution in this work focuses on input-related aspects, such as the linear combination of language embeddings and the alignment of visual observations. While I agree that these input designs can leverage the commonsense knowledge of the pretrained single-arm policies, this framework **overlooks the cooperation between the two arms**. In this design, the policies for the left and right arms are nearly independent of each other. Although this approach may be effective for certain bimanual tasks that assign fixed roles to each arm (e.g., one arm for stabilizing and the other for acting), it may not perform well in tasks requiring deeper collaboration between the arms (e.g., jointly lifting a large ball, where AnyBimanual achieves relatively lower scores). While I understand that such cooperation can be learned from data (i.e., the limited bimanual demonstrations), I believe it would be beneficial to explicitly ensure dual-arm cooperation within the framework design. In summary, I appreciate the efforts to transfer the single-arm policy to a dual-arm system, but there are still limitations regarding the cooperation of the two arms.
* In the last paragraph in Section 3.3, I agree with 'bimanual manipulation tasks often involves asynchronous collaboration that requires the left and right arm to attend to different areas of the whole workspace to act as different roles.' However, some tasks require both arms to work in the same small area, such as handing over a small item (as illustrated in Figure 5). In these cases, it is crucial to avoid collisions between the arms during the handover process; therefore, it would not be reasonable for the left arm to ignore the right arm.
* The authors are encouraged to conduct more real-world experiments on different tasks.

**Questions:**

* AnyBimanual exhibits relatively lower performance in tasks such as lifting a ball and sweeping into a dustpan. Could the authors provide further explanation for this?
* Typo: Line 96, "pol-icy" should be corrected to "policy."

---

### Official Review · Reviewer_kaUc · 2024-11-03

**Soundness:** 3
**Presentation:** 3
**Contribution:** 3
**Rating:** 3
**Confidence:** 4

**Summary:**

The paper propose a plug-and-play method named AnyBimanual, which transfers pretrained single-arm policy to multi-task bimanual manipulation policy with limited bimanual demonstrations. It first introduce a skill manager that dynamically schedules discovered skill primitives from the pretrained single-arm policy for language embedding boosting. Then it proposes a voxel editor to generate spatial soft
masks with for visual embeddings of the workspace aim to align visual input of single-arm policy model for each arm with those during pretraining stage.

**Strengths:**

+ High Practical Value: This is a very good idea and topic. Indeed, most of the current research focuses on single-arm operations. For dual-arm operations, the difficulty in obtaining demonstrations makes them less mature compared to single-arm operations. Therefore, transferring policies from single-arm to dual-arm operations is a very promising approach.
+ Intuitive and Reasonable Pipeline: The skill manager and voxel editor proposed in the paper align well with intuitive approaches to solving this problem and are designed reasonably.
+ Clever Design: The paper employs several ingenious designs, which are listed below:
   + Model Agnostic: The entire pipeline is model-agnostic, meaning it is independent of the specific single-arm pre-trained models used. As a result, it can be applied to a wide range of single-arm models.
   + Good design: The paper includes several ingenious designs, such as using KL divergence to guide the model in implicitly decomposing dual-arm workspace observations into two single-arm observations.

**Weaknesses:**

+ Lack of baseline: This paper includes only two baselines: one is PerAct, which is primarily designed for single-arm policies, and the other is PerAct-LF. Therefore, there is only one baseline that fully aligns with the setting of this paper. I suggest that the authors consider including more baselines for comparison. While I understand that there are fewer methods in the field that focus on dual-arm operations guided by language, the authors could still compare their work with pure dual-arm algorithms, such as DualAfford[1]. Since language-guided tasks are more challenging and this paper emphasizes cross-task generalization, the performance might not be as high as that of algorithms focused on cross-object generalization. However, these comparisons are essential and necessary to validate the effectiveness of the proposed method.
+ Generalization design: This paper makes significant efforts and design choices to preserve generalization ability. However, the final training method adopted undermines this goal.
     + Specifically, the paper uses an end-to-end training approach with a cross-entropy loss calculated between the predicted action and the final action (although there are other loss designs, this is the primary one). This approach makes the method more akin to imitation learning, which is known for its poor generalization capabilities. As a result, it seems to undermine the generalization ability that the pretrained single-arm model originally possessed.
     + Moreover, the paper still uses 20 or 100 demonstrations per task to train a model, which makes the voxel editor and skill manager task-specific and difficult to transfer to other tasks. For task-specific imitation learning, there are already well-established methods in the field, such as DP3.
     + Lack of Generalization Validation: As mentioned above, the experimental content lacks substantial validation of generalization. I recommend that the authors combine all tasks and demonstrations to train a single model and then directly test the model's generalization on new tasks. Such a training approach would hopefully enable the voxel editor and skill manager to learn cross-task knowledge and apply it directly to new tasks.
+ The design of pure implicit representation: The two main modules proposed in this paper are entirely implicit, which necessitates end-to-end supervision. For the skill manager, existing large models can explicitly address this task using language prompts. Therefore, we encourage the authors to include a comparison with this explicit approach using language prompts.

**Questions:**

+ Skill Manager: This section is not very clear. The skill manager seems to need to both learn primitive skills from demonstrations and decompose new instructions. Are these two tasks handled by a single module and, if so, how exactly it is trained. We recommend that the authors include a figure to illustrate this process.
+ Figure 2: Figure 2 is not clearly depicted, and the authors need to provide more explanation regarding the Mutually Exclusive Prior and the Sparsity Prior. Specifically, it should be clarified why these priors direct the training of the voxel editor and the skill manager. I recommend that the authors add more detailed explanations to make these concepts clearer.
+ RealWorld Experiment: I noticed that the simulation in the paper uses the Franka robot, while the real-world experiments use the UR5 robot. These two robots have significantly different workspaces and speeds. Although the authors use the end-effector (EE) pose, the differences in gripper length between the Franka hand and the Robotiq 85 gripper can lead to substantial sim-to-real gaps in these poses. The authors need to address How was the sim-to-real transfer performed. What methods were used to bridge the sim-to-real gap, especially considering the differences in gripper lengths and EE poses? In my opinion, author should first real2sim collected demonstration to simulator and change robot to train model then do sim2real and change robot again.

---

### Note · Authors · 2024-11-14

**Comment:**

Dear reviewers, thanks a lot for your time and effort in handling our manuscript! We really appreciate the valuable comments and questions from all the reviewers, as all of them are important to improve the quality of this manuscript. We are now working to conduct extensive experiments and rewrite the paper, including more compared baselines, more analysis, and more real-world results, to make it more valuable. Thus we would like to withdraw this paper at this moment, thank you very much.

**Withdrawal Confirmation:**

I have read and agree with the venue's withdrawal policy on behalf of myself and my co-authors.